# Microneedle Patterning of 3D Nonplanar Surfaces on Implantable Medical Devices Using Soft Lithography

**DOI:** 10.3390/mi10100705

**Published:** 2019-10-16

**Authors:** Sun-Joo Jang, Tejas Doshi, Jerusalem Nerayo, Alexandre Caprio, Seyedhamidreza Alaie, Jordyn Auge, James K. Min, Bobak Mosadegh, Simon Dunham

**Affiliations:** 1Dalio Institute of Cardiovascular Imaging, NewYork-Presbyterian Hospital and Weill Cornell Medicine, New York, NY 10021, USA; drjalive@gmail.com (S.-J.J.); tnd2001@med.cornell.edu (T.D.); aac2009@med.cornell.edu (A.C.); sea2012@med.cornell.edu (S.A.); jordynauge@gmail.com (J.A.); jkm2001@med.cornell.edu (J.K.M.); 2Department of Radiology, Weill Cornell Medicine, New York, NY 10021, USA; jdna2017@mymail.pomona.edu

**Keywords:** microneedle, micropattern, thermoplastic polyurethane, soft lithography, soft robotics, anchoring

## Abstract

Micropatterning is often used to engineer the surface properties of objects because it allows the enhancement or modification of specific functionalities without modification of the bulk material properties. Microneedle arrays have been explored in the past for drug delivery and enhancement of tissue anchoring; however, conventional methods are primarily limited to thick, planar substrates. Here, we demonstrate a method for the fabrication of microneedle arrays on thin flexible polyurethane substrates. These thin-film microneedle arrays can be used to fabricate balloons and other inflatable objects. In addition, these thin-filmed microneedles can be transferred, using thermal forming processes, to more complex 3D objects on which it would otherwise be difficult to directly pattern microneedles. This function is especially useful for medical devices, which require effective tissue anchorage but are a challenging target for micropatterning due to their 3D nonplanar shape, large size, and the complexity of the required micropatterns. Ultrathin flexible thermoplastic polyurethane microneedle arrays were fabricated from a polydimethylsiloxane (PDMS) mold. The technique was applied onto the nonplanar surface of rapidly prototyped soft robotic implantable polyurethane devices. We found that a microneedle-patterned surface can increase the anchorage of the device to a tissue by more than twofold. In summary, our soft lithographic patterning method can rapidly and inexpensively generate thin-film microneedle surfaces that can be used to produce balloons or enhance the properties of other 3D objects and devices.

## 1. Introduction

Micropatterning is a novel technique for engineering surface properties, such as adhesion [1] and wettability [2], and related biological processes and applications, such as inflammatory response [3] and cell manipulation [4,5]. It has been used in methodologies for testing drugs [6] and for the fabrication of a variety of devices within the fields of biochemical sensors [7], microfluidics [8], and organs on a chip [9,10]. Current micropatterning approaches are frequently limited to simple patterns and planar surfaces and can be time-consuming [4,8].

We previously developed a new approach for micropatterning 3D non-planar surfaces that combines low-cost soft lithographic embossing and vacuum bagging; however, only patterns with heights in the order of or smaller than the thickness of the film itself could be obtained [11]. In the study, micropatterns inspired by shark skin riblets and tree frogs were demonstrated in complex 3D surfaces. This technique is ideal for integrating the micropatterning technology for the rapid prototyping of various medical devices.

Microneedle arrays are among the most actively applied micropatterns, especially in drug and vaccine delivery studies [12,13]. Microneedles have been shown to have powerful drug delivery capabilities and improved patient safety and compliance [14,15]. Microneedle arrays were also applied as a novel type of adhesive mechanism for augmenting skin wound closure [1]. Microneedles based on the proboscis of the spiny-headed worm, which is solidly anchored to the fish intestinal wall, have been fabricated to mechanically interlock to the planar skin tissue.

There are various approaches to the fabrication of microneedle arrays [16]. Silicon microneedles are often made by photolithographic methods using deep reactive ion etching or wet etching in a bath of potassium hydroxide [17,18]. Laser cutting methods can also be utilized for the fabrication of microneedles [19,20]. Soft polymer microneedles have been often made by casting a polymer liquid solution into inverse molds [21]. However, most of these methods produce microneedle arrays with thick (>1 mm) planar substrates, where the microneedles possess mechanical properties that match those of the substrates themselves [1,22].

Here, we highlight a simple and straightforward approach to fabricating rigid microneedle arrays on flexible elastic substrates and the ability to easily use these films to create balloons or to apply these microneedle arrays to underlying flexible substrates. One advantage of this approach is that the films are fabricated on the basis on microneedle templates, thus any microneedle geometry can be utilized, provided that a silicone template can be molded in the required geometry. While a variety of other approaches demonstrate the ability to fabricate rigid microneedle arrays on flexible substrates, they typically fabricate rigid microneedles on the substrate using techniques such as 3D printing or electrodeposition [14,23,24,25]. While these approaches provide their own unique advantages (such as in the production of conductive microneedles), many require the microneedles to be serially fabricated on the films and/or need specialized tools or processes (e.g., x-ray lithography, electroplating, or reactive ion etching). Here, we present a simple and straightforward technique to fabricate rigid microneedle arrays on flexible substrates, which only requires basic tools for silicone molding and a spin coater.

## 2. Materials and Methods

### 2.1. Soft Lithographic Microfabrication of a Thin Flexible Microneedle Array

The microneedle fabrication process is shown in Figure 1a. We started from purchasing a commercially available polycarbonate microneedle array master mold (Smicna Pte Ltd, Singapore), which was fabricated by the proprietary version of the electro-discharge-machining (EDM) method [26]. The microneedle mold master used was characterized by a needle height of 300 µm, a needle width of 200 µm, a pitch of 2000 µm, and a total array size of 48 mm × 78 mm. The master mold was washed with isopropanol and dried with a nitrogen gun to remove any dust or contamination on the surface. A polydimethylsiloxane (PDMS) solution was prepared by mixing SYLGARD^TM^ 184 Silicone Elastomer base and curing agent (Dow, Auburn, MI, USA) at a ratio of 10:1. PDMS was cast on the polycarbonate microneedle master, degassed, and cured in an oven (110 °C, 30 min). The PDMS mold was then detached from the polycarbonate master.

The PDMS inverse mold (typically, 5–6 mm thick) was cleaned with isopropanol and dried in the clean room. A medical-grade thermoplastic polyurethane (TPU) solution was prepared by mixing 15 g of Tecoflex^TM^ SG-60D (Lubrizol, Wickliffe, OH, USA) and 100 g of dimethylacetate (13.8 wt% Tecoflex solution) with the help of a centrifugal mixer (2000 rpm, total 12–18 h). The TPU solution was then spin-coated on the inverse mold (1800 rpm for 60–90 sec) and incubated in an oven (80 °C for 30–45 min). The first TPU layer was gently cleaned with isopropanol, smeared with a UV curable resin solution (Norland Optical Adhesive 63, Norland Products, Cranbury, NJ, USA), and then degassed. The excess resin was removed by forceful blading using glass slides. The resin was then cured by 365 nm UV light exposure with irradiance of 500 mW/cm^2^ for ~10 seconds (LC-L1V3, Hamamatsu Photonics K.K., Hamamatsu, Japan). Next, a second Tecoflex TPU layer was spin-coated (400 rpm for 60–90 sec) and cured in the oven (80 °C for 30–45 min), encapsulating the rigid microneedles between two layers of TPU. The final microneedle array membrane was gently peeled off the inverse mold. High-quality films were consistently achieved.

One advantage of this method is that it can be applied to microneedle arrays based on any template. To illustrate its generalizability, we used a method to create alternative templates based on laser-cutting sheets of acrylic, described in detail elsewhere [27]. These templates could also be effectively utilized to produce high-quality microneedle films (Appendix A).

### 2.2. Transfer of Microneedle Arrays to Other Objects

Thin-film microneedle arrays were attached to the surface of a prototype patient-specific left atrial appendage occlusion device [28] and to the fingertips of a glove, through the use of soft lithographic vacuum bagging, whereby the object is wrapped in the microneedle array film and then placed in a vacuum bag at elevated temperatures until the polyurethane film is thermally bonded to the object. This process is described in detail elsewhere [11].

### 2.3. Rapid Prototyping of Microneedle-Patterned Soft Robotic Medical Device

We applied microneedle-patterning to a proof-of-concept soft robotic implantable polyurethane stent (Figure 2) [29,30]. The fabrication of the stent itself is discussed in further detail elsewhere [29] and only briefly here: The TPU sheets and water-soluble polyvinyl alcohol (PVA) films were laser-cut (VLS 2.30, Universal Laser Systems, Scottsdale, AZ, USA) into hexagonal-hole arrays. The stent was designed to have a 19 mm diameter based on the size of the typical porcine aorta. The PVA layer was designed to create the inflatable regions of the balloon, once sandwiched between two layers of TPU film. The TPU–PVA–TPU layers were then heat-pressed at a temperature of 136 °C and with a compression force of 25 kN for 30 min. A stretchable UV-curable adhesive (Dymax 1165-M) was used to attach the non-needle surface of the flexible microneedle array to the outer surface of the polyurethane stent by applying UV light with irradiance of 500 mW/cm^2^ for 10 seconds (LC-L1V3, Hamamatsu Photonics K.K., Hamamatsu, Japan). The planar stent was then rolled into a cylinder, and its edges were glued together to maintain this shape. The PVA layer was dissolved by injecting water into the inlet of the inflatable stent, using a syringe. Polyurethane stents without microneedles were also created as a control.

### 2.4. Anchorage Test of the Microneedle-Patterned Polyurethane Stent

We performed anchorage tests of the microneedle-patterned polyurethane stents using a columnar tensile testing system (Instron, 5943, Norwood, MA, USA). Porcine aortas were extracted and cleaned by removing outer fat tissues. The polyurethane stents were placed inside the aorta and pressurizedby injecting the required amount of water (inflation volume of 1.5 mL). The distal end of the aorta was fixed into the lower gripper of the testing system. Custom-built tethers were applied to the stent and connected it to the top gripper of the tensile tester. We used a tensile setup with load cell capacity of 250 N, force measurement accuracy of ± 0.00125 N, and displacement accuracy of ± 0.02 mm. We continuously recorded the force applied to the load with constant extension rate (1 mm/min, 15 minutes). Control stents without the microneedle pattern were also tested in the same setting. After the pressurized implantation of the microneedle-patterned stent and pull-out testing, we checked the integrity of the microneedles on the stent surface under a stereo microscope.

### 2.5. Mechanical Integrity Test of the Microneedle Film

To understand the nature of the interface between tissue and microneedle films, films were studied under a stereoscope. Microneedle arrays were placed in contact with square sections of porcine aorta. A glass slide was utilized to apply a downward normal force pushing the microneedle film into contact with the aorta. Meanwhile, a lateral force was applied to the film and gradually increased until microneedle slip was observed (Appendix A).

### 2.6. Imaging

The microneedle array was observed by stereomicroscopy and scanning electron microscopy (SEM) (Helios Nanolab 660 SEM/FIB, FEI, Hillsboro, OR, USA). Volumetric measurements of the microneedle were performed using laser scanning confocal microscopy (LSM 700, Carl Zeiss, Oberkochen, Germany) to evaluate the uniformity of the microneedles. The autofluorescence signal from the microneedles could be detected using a 488 nm excitation laser and a 518 nm emission filter. The microneedles were observed with a plane apochromat 20× lens (numerical aperture, 0.8), using a field of view of 320 µm × 320 µm and a section thickness of 5 µm. The height, base width, and tip sharpness of the microneedles were measured. Confocal microscopy of the microneedles with and without pressurization was also performed.

### 2.7. Statistics

All analyses were performed using R software, version 3.5.3 (R Core Team, 2019). For the comparison of maximum load, the unpaired t-test was performed with two-sided *p* values and *p* values < 0.05 as statistically significant. Error bars shown in the graphs indicate the standard deviation.

## 3. Results

Figure 1a shows the workflow for the fabrication of thin-film microneedle arrays. Microneedle arrays with TPU–NOA63–TPU layers were gently peeled off from the inverse mold (Figure 1b). Stereomicroscopy and SEM confirmed that the microneedles were sharp and uniformly spaced on the thin flexible TPU membrane (Figure 1c,d). The thickness of the thin TPU membrane fabricated with our protocol was 53 ± 7 µm (n = 8). This simple technique is generalizable to many applications. Figure 3 shows confocal microscopy images of an individual microneedle. The mean height of the microneedle was 289 ± 10 µm (n = 15), and the mean width of its base was 195 ± 10 µm (horizontal) and 215 ± 10 µm (vertical) (n = 15). The radii of curvature at the tip of the microneedle were 4 ± 3 µm (horizontal) and 4 ± 2 µm (vertical) (n = 15). The analysis of microneedles fabricated from an alternate template, using a laser-cutting method, is available in Appendix A.

Figure 4 shows images of both the glove and the occluder following microneedle array transfer. The microneedle films could be employed as thin films (Figure 4a). Additionally, they could be shaped into balloons or attached to other complex 3D objects. Figure 4b shows the microneedle array attached to the 3D nonplanar surface of an implantable left atrial appendage occlusion device, which was manufactured in a patient-specific design. As another application, the microneedle array was also attached on the fingertips of a glove (Figure 4c).

To test the role of microneedle patterning in enhancing the anchorage of implantable medical devices, we fabricated a soft robotic inflatable polyurethane balloon stent that was developed for the percutaneous treatment of aortic dilatation (Figure 5). The outer diameter of the stent was 19 mm, which was matched to the inner diameter of the porcine aorta sample. A thin flexible microneedle membrane was successfully attached to the outer surface of the polyurethane stent (Figure 5a,b). After pressurizing the balloon of the stent with water, the stent was expanded on both the inner and the outer surfaces with a honeycomb pattern (Figure 5c,d). The microneedles were well adherent to the surface, maintaining their positions on the stent (Figure 5e,f). The change in morphology of an individual microneedle, before and after inflation, was characterized by confocal microscopy and found to be negligible (Appendix A).

We tested the anchorage of the microneedle-patterned polyurethane stent in a pressurized state (Figure 6). As the stent was pulled-out vertically from the fixed porcine aorta, the force applied to the load increased gradually until it reached a peak value and then decreased continuously. The load from a microneedle-patterned stents was higher than that from a control stent throughout the 15 mm of the extension. The maximum load measured during the constant extension test was 1.3 ± 0.2 N for microneedle-patterned stents (n = 8) and 0.5 ± 0.1 N for control stents (n = 8) (*p* < 0.001). The microneedles on the stent remained intact after the pressurized implantation and pullout test (Appendix A).

No mechanical failure of the film or the individual microneedles was observed during the lateral pull-out test under normal compression (Appendix A). The microneedles appeared tilted during the pullout test but stayed attached to the array film without breaking. Stereoscopy of the microneedle array after the experiment showed intact and sharp microneedles over most of the array. The inner wall of porcine aorta remained intact without indents, punctures, or erosions in the microneedles (Appendix A).

## 4. Discussion and Conclusions

In summary, we developed a soft lithographic micropatterning technique to rapidly and inexpensively create thin films of microneedle arrays and used them to micropattern the surface of balloons, work gloves, and implantable medical devices. This method is simple, straightforward, and presents advantages for the fabrication of emerging medical devices with patient-specific designs. As a proof of concept, a rapidly prototyped inflatable polyurethane aortic stent was micropatterned with the microneedle array. We also showed that the anchorage of the microneedle-patterned surface was 2.6 times better than that of a non-patterned surface.

The PDMS mold could be repetitively used to generate multiple ultrathin flexible TPU polymer microneedle arrays. We used Tecoflex TPU, which has been used in many implantable devices due to its biostability, processability, and biocompatibility [31,32]. The microneedle arrays fabricated here were made from a UV-cured polymer (NOA 63) with modulus of elasticity of 240,000 psi, tensile strength of 5000 psi, and Shore D hardness of 90 [33]. I These needles appeared stiff and sharp enough to engage the tissue without bending or failing, thus providing enhanced anchoring, but not so stiff or sharp as to puncture and damage the underlying tissue.

There are many other cost-effective and scalable approaches for making flexible microneedle arrays. The 3D printing technology provides a promising approach. M. Luzuriaga et al. showed microneedle array fabrication using Fused Deposition Modeling (FDM) 3D printing [14]. Economidou et al fabricated a 3D-printed microneedle master using stereolithography (SLA) [23]. However, these methods can only produce needles whose sharpness is limited by the resolution of the printer. By using a prefabricated template with sharp microneedle features, the approach presented here allows the fabrication of sharper microneedle arrays, along with the application of rigid microneedles on flexible films. We believe that future improvements in 3D printing technology would lead us to a simple and scalable approach for the fabrication of flexible and scalable microneedle arrays [34].

A variety of micropatterns have been demonstrated to increase anchorage forces. We previously reported “Tree-frog”- and “Shark skin riblets”-inspired micropatterning for nonplanar surfaces of 3D devices [11]. Those patterns were engineered for enhancing wet friction and fluid shear stress on the nonplanar surface of implantable medical devices. Yang et al. showed the feasibility of improving the anchorage to a planar surface with a swellable microneedle pattern [1]. In this study, microneedle patterns were engineered to enhance anchorage through the application of a normal force on the implanted tissue. Anchorage is enhanced when the implantable balloon is inflated and the biological tissue is pushed normally to the surface of the device. This approach can be applied to various medical devices (left atrial appendage occluders, vascular aneurysm occluders, soft robotic vascular stents, etc.) which use balloon inflation for their implantation. To clarify the novelty of our approach, several microneedle fabrication methods were compared in Appendix A.

Future studies will analyze the effect of different heights, aspect ratios, and pitches on the anchoring force vs the flexibility of the film for various nonplanar medical devices, the impact on the immune and inflammatory responses, and the long-term stability of the microneedles and the devices through in vivo experiments.

## Figures and Tables

**Figure 1 micromachines-10-00705-f001:**
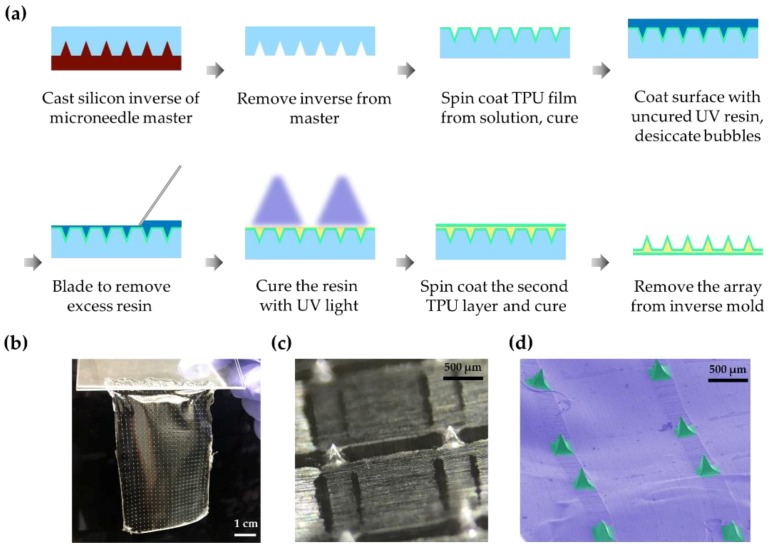
Fabrication of the microneedle array. (**a**) Schematic of the microneedle array fabrication; (**b**) Thin microneedle array peeled from the inverse mold; (**c**) Stereoscopic image of the magnified microneedle array; (**d**) Scanning electron microscopy of the microneedle array (pseudo-colored). TPU: thermoplastic polyurethane.

**Figure 2 micromachines-10-00705-f002:**
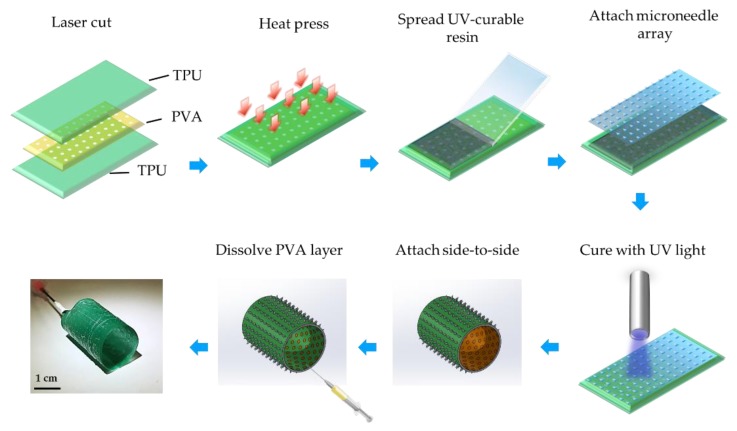
Schematic for microneedle patterning in a rapid prototyped soft robotic polyurethane stent. TPU sheets were laser-cut into a stent design matching porcine aorta. Water-soluble polyvinyl alcohol (PVA) films were laser-cut to obtain an array with hexagonal holes. The combined layers were then heat-pressed (136 °C, 25 kN for 30 min). The top TPU layer was covered with a UV-curable adhesive and attached to the microneedle array. The adhesive was then cured by short exposure to UV light. The planar stent was then rolled into a cylinder, and its edges were glued together. The PVA layer was then dissolved in water using a syringe.

**Figure 3 micromachines-10-00705-f003:**
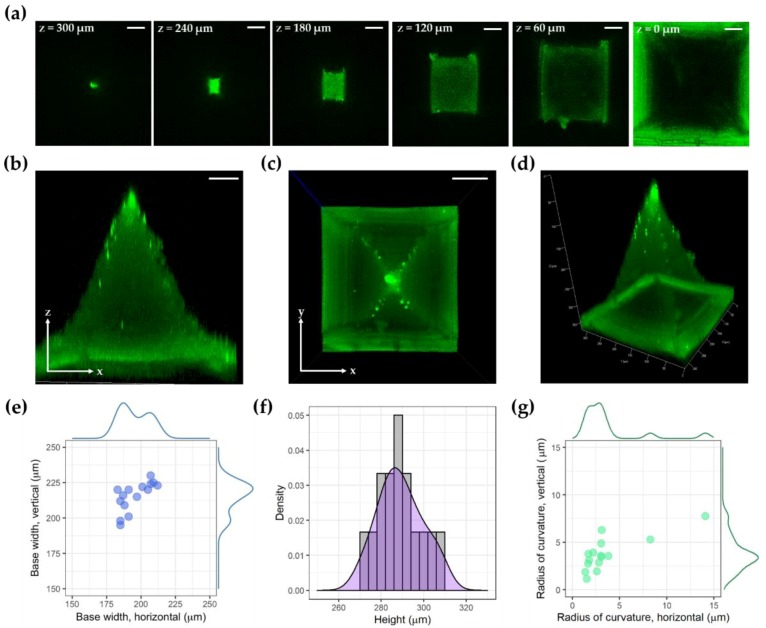
Confocal microscopy imaging of microneedles. (**a**) Cross-sectional images of microneedles with XY cross section at z = 0, 60, 120, 180, 240, and 300 µm; (**b**) Maximum intensity projection (MIP) image by y-directional projection; (**c**) MIP image by z-directional projection; (**d**) MIP image from arbitrary angle; (**e**) Scatter plot for horizontal and vertical widths of the microneedle base; (**f**) Histogram of microneedle heights; (**g**) Scatter plot for horizontal and vertical radii of curvature. Scale bar, 50 µm.

**Figure 4 micromachines-10-00705-f004:**
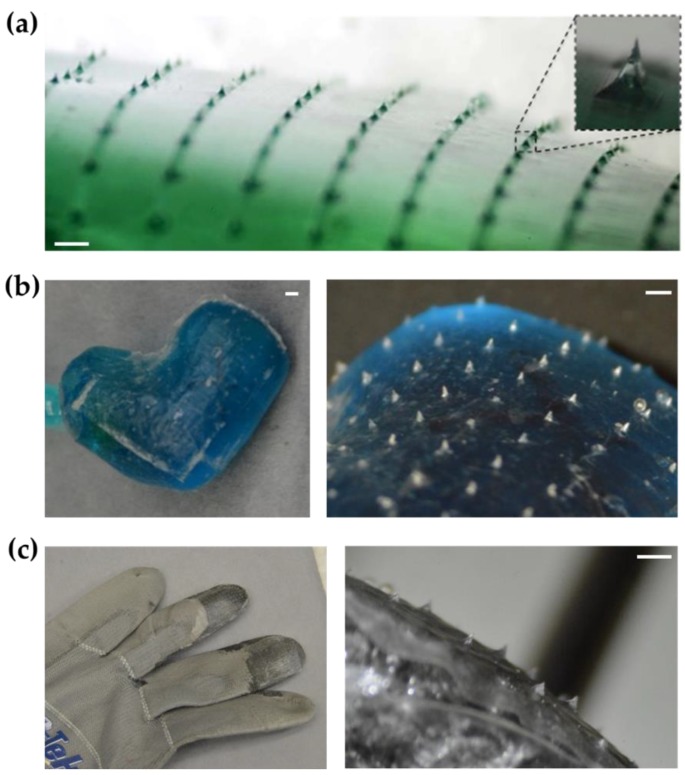
Examples of microneedle patterning on nonplanar surfaces. (**a**) Wide curved surface; (**b**) Left atrial appendage balloon occlusion device with patient-specific design; (**c**) High-friction glove. Scale bar, 1 mm.

**Figure 5 micromachines-10-00705-f005:**
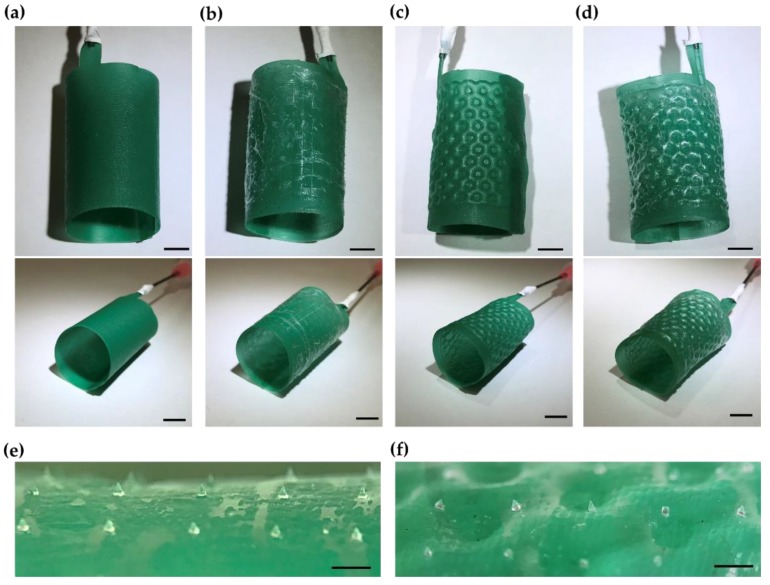
Microneedle-patterned soft robotic polyurethane stents. (**a**) Unpressurized stent without surface patterning; (**b**) Unpressurized stent with microneedle patterning; (**c**) Pressurized stent without surface patterning; (**d**) Pressurized stent with microneedle patterning; (**e**) Magnified view of the microneedles, unpressurized; (**f**) Magnified view of the microneedles, pressurized. Scale bar, 5 mm (**a**–**d**) and 1 mm (**e**,**f**).

**Figure 6 micromachines-10-00705-f006:**
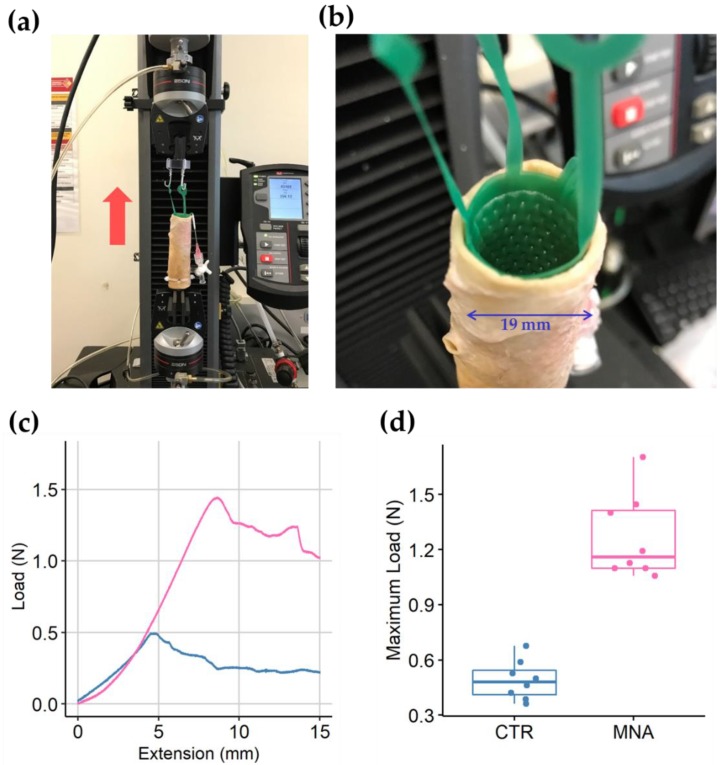
Anchorage tests for microneedle-patterned stents in a porcine aorta. (**a**) Tensile test setup on the Instron machine with vertically located aorta and stent; (**b**) Pressurized stent with linkers for the extension test; (**c**) Plot of continuously measured load during 15 minutes pull-out at 1 mm/min extension rate; (**d**) Box plot for maximum load applied during the test on control (CTR) and microneedle-attached (MNA) stents.

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
