# Peer review of "Microneedle Patterning of 3D Nonplanar Surfaces on Implantable Medical Devices Using Soft Lithography"

_micromachines, 2019, doi:10.3390/mi10100705_

Round 1

Reviewer 1 Report

The authors have outlined a simple method of transferring MN textures to the surface of non-planar objects, and have undertaken limited mechanical testing to demonstrate an increase in friction.

Background information is well presented and methods and materials are detailed and specific.

My main concern with this paper is the limited testing undertaken on the impact of side force on MN integrity and strength. Will these break off in tissue? Will the MN cause damage to the tissues in which they are in contact? What impact on immune response or inflammatory response would be expected? How can the authors demonstrate that the MN layer will not detach from the non-planar surface? What would the expected lifetime be of such a system? The authors should provide information on the uniformity of MNs produced, for example geometry, height, base width, tip sharpness.

Reviewer 2 Report

This paper tackles the fabrication of microneedle (MN) structures on flexible substrates using micromolding. The method of fabrication enables flexible MN structures that can eventually be transferred to medical devices and an example is illustrated briefly in this work.    

Technical comments:

Title: The title is a bit misleading – you don’t really pattern in 3D surfaces. You fabricate a 3D device and attach it to non-planar substrates. Abstract: MNs have been fabricated on flexible surfaces. For example - Ren et al., MPDI Sensors, 2018 (https://doi.org/10.3390/s18041191). How would you differentiate from such work? Intro: 3D Printed MNs (several examples including S. Economidou et al., Mat. Sci. and Eng C, vol. 102, pp. 743-55, 2019; M.Luzuriaga et al., Lab on a Chip, 2018, 18, 1223). How do you differentiate from 3D printing? It is an effective way to micropattern rapidly and cost effectively. I believe even flexible 3D printed structures have been demonstrated. Intro: The variety and scope of microneedle fabrication technologies not to mention flexible MN fabrication has not been treated at all in this intro. The authors are referred to recent reviews such as Prausnitz, Annu. Rev. Chem. Biomol. Eng. 2017. 8:177–200 and others. This reviewer would recommend treating MN fabrication technologies that are numerous over the years and try and bring out the uniqueness of their approach in the Intro section. This is not clear at all in the current version of the paper. Materials and Methods: How was the mold master fabricated? Any details or could you point to a reference? Materials and Methods: What are the feature sizes on the mold master? Normally array sizes, pitches, heights etc. are defined in this part. Also, how well are these sizes transferred to PDMS? Have you performed and design to device? This could be a good result that can be added to the paper. Materials and Methods: Any separation layers needed? Typically, polymers of low surface energies need separation layers to pull one from the other during micromolding. What are the surface energies of the two materials? Are they easy to separate? How many times have you performed this process? How repeatable is it from the perspective of feature definition? Figure 1: Scale bars for all smaller figures Materials and Methods: How thick was the PDMS layer? How did you spin coat this layer? Normally spin coating needs a solid surface and PDMS is flexible and notoriously hard to spin coat on. Materials and Methods: UV exposure: Lamp or mask aligner? Energy used? Materials and Methods: Section 2.2 - This section needs more info: Are there any images of the final device? If so, please include them in the paper. A schematic of how the process is executed will help. What is the mechanism of bonding? Have you tested the bond strength? Section 2.3: What laser? What sorts of patterns? Section 2.3: This is a fabrication journal. Details on designs of the device and detailed fabrication are necessary. What was the size of the stent? You are not patterning on the stent but applying the MN to the stent. This is not unique. If there was patterning on non-planar surface that would be novel. Figure 3: Figure 3 needs a legend. Also, in the last image which appears to be a stereoscopic image, we need a close up of the MN. From this image it is unclear what the structure is. Also, please include scale bars. Results: Quantification of tip sharpness and radius of curvature (ROC)? Results: What is “Waist”? Is that Width? What are some of the other configurations that the MNs can be fabricated in? Results: Fig. 2 comes after Fig. 3. You need to fix this. What does thin film configurations mean? In microfabrication thin films are in nanometer scale thickness usually. Figure 2: Needs scale bars and also some higher quality images. Are there SEM images of the MNs? Figure 4: Needs scale bars and also how are the dimensions of the MNs before and after being pressurized? Need some quantification here other than qualitative images? Also, how repeatable is this step? Results: Most MN papers have a lot more characterization: Do we have images of the porcine aorta where we can assess whether the MNs are making indents and how repeatable are these indents? What are the sizes of the indents? What about yield strength of the MNs? Can you quantify that? Soft material MNs need yield strength quantification and comparison with other hard material MNs. Other results usually present in MN papers is some application - you have shown anchoring but is this a long term application. Maybe the MNs can be versatile and used as a wearable device? Can you show some feasibility here?

Round 2

Reviewer 2 Report

This paper tackles the fabrication of microneedle (MN) structures on flexible substrates using micromolding. The method of fabrication enables flexible MN structures that can eventually be transferred to medical devices and an example is illustrated briefly in this work.    

Technical comments:

The authors have done additional work as requested and have addressed many of the concerns raised by the reviewer. A couple of major issues still need to addressed:

The major justification and the only result shown appear to be anchoring and not utilization of traditional MN structures for applications such as drug delivery or biopotential measurement. Why does anchoring need MN structures? Couldn't you fabricate "gecko-feet" type structures and in that way, you can completely move away from MN based comparisons? Could something like that be fabricated using this process? Can you demonstrate the quantitative anchoring differences between MN and "gecko-feet" type structures? Scalability and cost effectiveness has been used numerous times to justify novelty of the work. While I certainly like the academic community bringing these terms into the novelty equation, perhaps a calculation of how cost effective this methodology is can enhance this paper tremendously. Could you do a cost calculation for this method compared to standard methods for MN fabrication such as 3D Printing, DRIE, laser, micromolding etc. A table of the results would be useful with references. I still think that if you are wanting to advertise this work as MN fabrication, additional characterization (mechanically typically) is necessary. For instance as suggested, What about yield strength of the MNs? Can you quantify that? Soft material MNs need yield strength quantification and comparison with other hard material MNs.

Minor issues:

Lines 53-55 in the intro seems incorrect to me. Can you justify with some references? Lines 64-67: Not really true. You do have an inverse EDM mold as well that you start with. EDM is a specialized process that not many have access to such a process step to make these molds. Figure 2 is really nice. Formatting of the figure headings seems to have inconsistencies. Please check. Line 110: Ref 26 is not work from this group. Have you used a laser micromachined template for MN fabrication? What were the results? Did you get something similar to Fig. 2? Can you add to the supplementary (a similar figure to figure 2). In Figure 5, I don’t believe the comment on quantitative characterization of unpressurized and pressurized MNs was addressed. I may have missed it but it does appear that there is a structural change from the image. What is this change? Figures need to be reorganized. Conceptual schematics and methods should come first and results should have figure numbers beyond these. Currently the figure organization looks odd. Line 225: "Slightly" tilted needs quantification - to what degree were they tilted? Lines 240-241: References for hardness and modulus of elasticity for NOA 63. Lines 248-49: Look at the sharpness of this recently published 3D printed MN paper (Johnson et al., PLOS One DOI:10.1371/journal.pone.0162518). How will you differentiate from something like this?

Round 3

Reviewer 2 Report

The authors have addressed some of the issues raised and have removed some text to address other issues raised by the reviewer. 

The reviewer still feels this paper could use further quantification of MN properties as suggested and some comparisons to other techniques of MN fabrication (a table as suggested). I believe these would enhance the paper's novelty since it is rather limited in its current form. 
